# A partner-driven decision support model to inform the reintroduction of bull trout

Joseph R. Benjamin[1]*, Judith Neibauer[2], Hugh Anthony[3], Jose Vazquez[4], Ashley Rawhouser[3], Jason B. Dunham[5]

1 U.S. Geological Survey, Forest and Rangeland Ecosystem Science Center, Boise, Idaho, United States of America, 2 U.S. Fish and Wildlife Service, Wenatchee, Washington, United States of America, 3 National Park Service, North Cascades National Park, Sedro Woolley, Washington, United States of America, 4 U.S. Fish and Wildlife Service, Mid-Columbia Fish and Wildlife Conservation Office, Leavenworth, Washington, United States of America, 5 U.S. Geological Survey, Forest and Rangeland Ecosystem Science Center, Corvallis, Oregon, United States of America

* jbenjamin@usgs.gov

## Abstract

Assessments of species reintroductions involve a series of complex decisions that include human perspectives and ecological contexts. Here, we present a reintroduction assessment involving bull trout (*Salvelinus confluentus*) using a structured decision-making process. We approached this assessment by engaging partners representing public utilities, government agencies, and Tribes with shared interests in a potential reintroduction. These individuals identified objectives, decision alternatives, and ecological scenarios that were incorporated into a co-produced simulation-based model of potential reintroduction outcomes. The model included mathematical representations of habitat availability, life history expression, and assumptions regarding constraints on potential bull trout populations. Within each recipient stream, partners chose to explore a wide range of decision alternatives and simulated scenarios affecting reintroduction success. Results suggested that 1) reintroductions using eggs or adults were most optimal, 2) adding more individuals resulted in diminishing returns, 3) access to migratory habitat could improve success, and 4) the diversity of opportunities for life history expression led to improved reintroduction opportunities. In addition, modeled scenarios indicated some recipient streams consistently produced lower abundance of reintroduced bull trout. This work contributes a novel example to a growing portfolio of reintroduction assessments that may inform future conservation for bull trout and many other species facing similar challenges.

## Introduction

Reintroduction is one conservation tactic that has been used worldwide to broaden distribution or restore threatened or endangered species [1]. However, evaluating the feasibility of reintroductions is not straightforward [2]. In part, this is owing to

**Data availability statement:** Simulated model results are available in ScienceBase (https://doi.org/10.5066/P14BQGBG).

**Funding:** Funding was provided by U.S. Geological Service-National Park Service Natural Resources Preservation Program to JRB, JBD, and AR. The funding program did not play any role in the study design, data collection or analysis, decision to publish or preparation of the manuscript.

**Competing interests:** The authors have declared that no competing interests exist.

uncertainty of how reintroduced species will respond both biologically (e.g., survival, reproduction) or ecologically (e.g., influences of environmental conditions). Furthermore, there are many questions related to how the reintroduction itself is implemented. For example, common questions may include, but not limited to, release location, number to release, life stage to release, and likelihood of success, as well as potential consequences to the donor population and recipient ecosystem. Furthermore, social considerations are just as important as ecological and environmental factors because multiple decisionmakers have different jurisdictional boundaries, perceptions, and beliefs of the best management decision [3].

One approach to addressing the challenge of assessing reintroductions is decision analysis and the process of structured decision making [4,5]. A critical part of this process involves engagement of managers and other practitioners (henceforth partners) to work together to explicitly identify the problem, objectives and means to reach identified objectives. In addition, structured decision making quantifies the values and beliefs of partners and uncertainty in biological and ecological dynamics within a modeling framework [5]. The rationale for using structured decision making is to deconstruct a complex problem into simpler steps, which includes 1) identifying the problem statement, spatial extent and fundamental objectives, 2) identifying realistic management alternatives to achieve the objectives, 3) assembly of a model of the system behavior, 4) running the model to evaluate outcomes under different management scenarios, and 5) evaluating sensitivity of the model to assess model and decision uncertainty. Partners were involved in discussions in each of these steps. Structured decision making has been used in many natural resource applications to aid in decision making [6–8], including reintroductions [9,10].

We applied a structured decision making approach to a reintroduction assessment for bull trout (*Salvelinus confluentus*). Bull trout is a threatened cold-water species whose distribution has been reduced and populations locally extirpated. Reasons for the decline include influences of habitat loss and fragmentation, wildfire, influences of introduced species, and climate change, among other threats [11,12]. Consequently, bull trout have been extirpated from much of their natural range, but in many locations past threats may no longer be acting, and there are potential opportunities for reintroductions. In the Lake Chelan watershed, WA (Fig 1), for example, bull trout were historically abundant [13]; however, by 1957 they were assumed to be functionally extinct. Potential drivers of this extirpation include overfishing, introduced disease, and intense flooding events that could have scoured spawning areas. Recently, partners have discussed reintroduction of bull trout into the historical habitat within the Lake Chelan watershed because many of the suggested causes of the initial extirpation may no longer be in effect (e.g., overfishing).

Here we engaged multiple partners from Tribal, public utilities, and governmental agencies to co-produce a model that explores alternatives for reintroductions of bull trout in the upper Chelan basin. Our overall objectives for this study were to 1) engage partners through a structured decision-making process, 2) document the process, and 3) communicate the results to inform future decisions to implement a

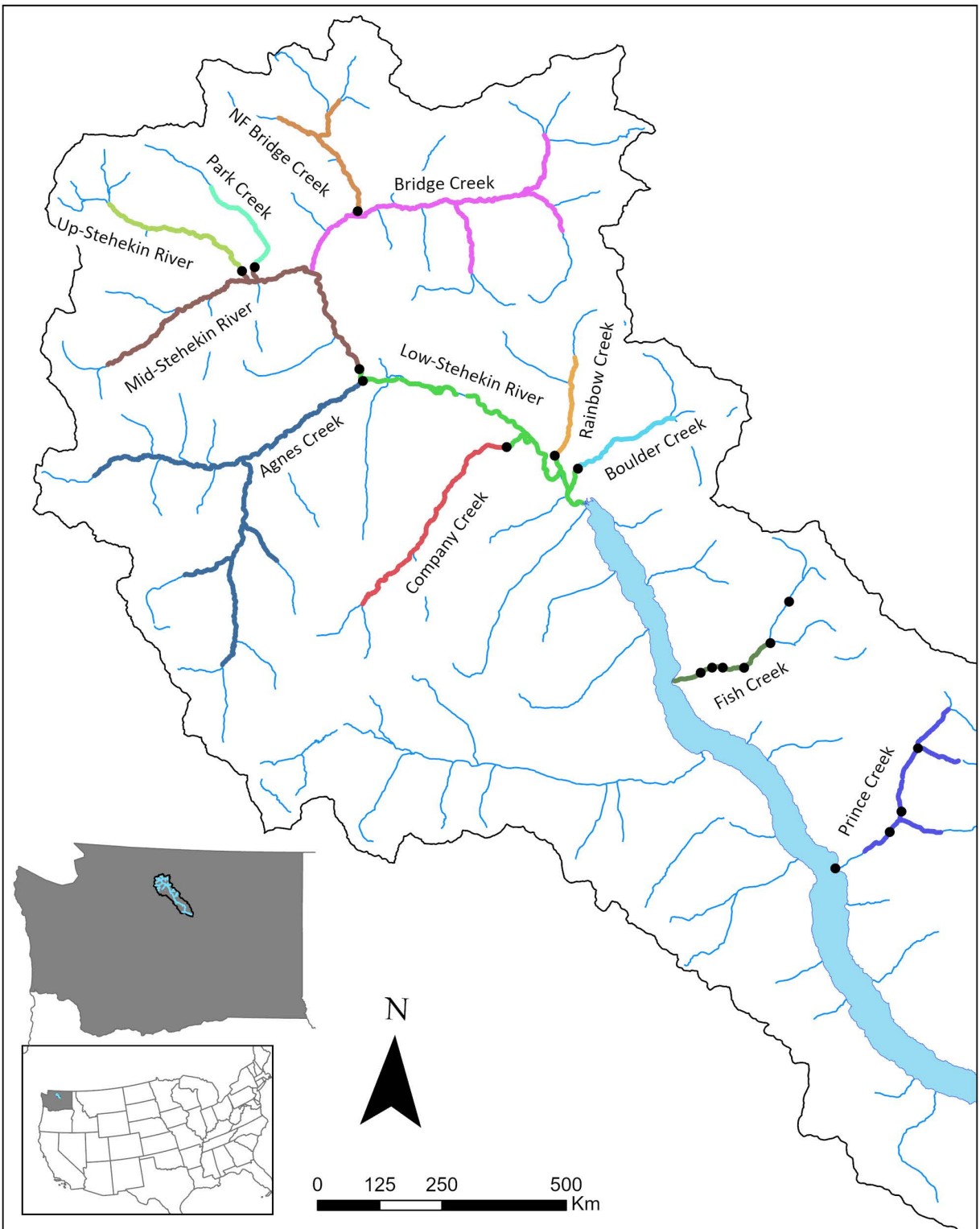

**Fig 1. Lake Chelan watershed and the twelve streams identified as potential recipient streams for bull trout reintroduction.** The highlighted portion of the stream represents the amount of suitable spawning and rearing habitat under current conditions. Black circles represent potential barriers to bull trout movement. Map was created in ArcGIS Pro (ESRI, Redlands, CA USA) using flowlines from U. S. Geological Survey [35] and state boundaries from U.S. Department of Commerce, Census Bureau, Cartographic Boundary Files.

reintroduction of bull trout and an example that others may follow. Findings from this effort can directly inform reintroductions within the area we considered, but also provide a useful case study among a growing number of formal assessments of reintroductions of bull trout across the species' range [14–19].

## Methods

We used a decision support framework centered around the process of structured decision-making [4,5] to evaluate the feasibility of reintroducing bull trout into the Lake Chelan watershed. Below we describe the steps in this process as it was applied. We engaged partners from federal, state and county governments, Tribes and county utility companies in monthly workshops to advise each step in the process. The process took approximately 20 months from January 2022 through August 2023.

### Study area

The Lake Chelan watershed lies partly in the North Cascades National Park, Washington State (Fig 1). Lake Chelan is the third deepest lake (453 m) in the U.S [20] A major inflow into Lake Chelan is the Stehekin River, the majority of which is protected within federal wilderness (U.S. Forest Service) and National Park Service lands. The watershed is notable for an abundance of available cold water fed by glaciers and a deep persistent snowpack. Water temperatures below 16°C are critical for bull trout [11], and current summer water temperatures in the lower Stehekin River average 11°C. Temperatures are predicted to remain below 15°C for at least the next 60 years [21].

Historically, bull trout were abundant in the basin and even supported a commercial fishery [13,22]. The last verified observation of bull trout in the Lake Chelan watershed was in 1957. Since that time, numerous surveys have been completed using a variety of methods including gillnets, snorkeling, electrofishing, and basin-wide environmental DNA [23–25]. Findings from this work indicate that bull trout have been extirpated throughout the watershed. The Lake Chelan watershed is considered historical core habitat by the U.S. Fish and Wildlife Service [11], and reintroductions have been identified as one action for potential recovery of bull trout.

Several native fish of interest occur in the system including Westslope cutthroat trout (*Onchorhynchus clarkii lewisi*), burbot (*Lota lota*), mountain whitefish (*Prosopium williamsoni*), pygmy whitefish (*P. coulteri*) and several sucker species (*Castostomus, spp.* [26]). Many nonnative fishes have been introduced including brook trout (*S. fontinalis*) and rainbow trout (*O. mykiss*) in the Stehekin River and tributaries and kokanee (*O. nerka*) and lake trout (*S. namaycush*), and Chinook salmon (*O. tschawytscha)* into Lake Chelan.

### Decision statement and objectives

The decision situation to summarize the purpose of the feasibility study, as agreed upon by the partners, was stated as: "Bull trout are functionally extirpated in the Lake Chelan watershed above Lake Chelan Dam creating a need to explore options for reintroduction to support distribution and conservation of bull trout within the upper Mid-Columbia Recovery Unit."

With the problem statement identified, fundamental objectives were discussed. The primary fundamental objective identified was to maximize the number of adult bull trout in a recipient stream. Minimizing the impact of reintroductions on other native fish was also discussed as a possible fundamental objective but was ultimately not included since it did not capture the intent of the problem statement. Similarly, the decision around the use of source populations for a reintroduction was heavily discussed and a major concern because some potential source populations in the Recovery Unit currently have low population abundances. Assessing the consequences of removing fish from a specific source population would require information on specific local populations (and thus considerable effort). Thus, this objective was classified as an action that would be addressed following this reintroduction assessment.

## Reintroduction alternatives

Reintroduction alternatives involve three parts; the recipient stream (Fig 1), life stage of bull trout to reintroduce, number of individuals to reintroduce, and the timing and duration of reintroductions of these individuals into the recipient stream. Twelve recipient streams or basins in the Lake Chelan watershed were selected to be considered for reintroductions (Fig 1). Delineation of streams was mostly based on potential barriers to upstream movement by bull trout. Because of this, some streams or parts of streams were combined. For example, the lower Stehekin River included the lower most section of Rainbow, Boulder, Company and Agnes Creeks that were below putative barriers. In contrast, partners were uncertain if potential barriers in Fish or Prince Creeks would prevent upstream movement of bull trout and chose to consider all potential habitat identified by local experts as a single contiguous unit.

Partners identified eggs, juveniles, subadults, and adults as the potential life stages that could be used for reintroduction. Instead of identifying a specific number of individuals per life stage, partners wanted to explore a range of options (none to high) to assess the point of diminishing returns as more individuals were reintroduced. For eggs, the range was 0–30,000 individuals added per recipient stream, and for juveniles the range was 0–4,000. For subadults and adults, the range was 0–300. Although specific methods for each stage could be identified (e.g., egg boxes for eggs or artificial propagation for juveniles), partners chose leave this decision until potential implementation occurs. We assumed the number of individuals would be added for five consecutive years and then stopped. For example, the first five years of simulations with 30,000 eggs would be added to a recipient stream each year and then fundamental objectives assessed for years 10, 30, 50 and 70.

## Decision support model description

Bull trout can exhibit partial migration where individuals may be migratory (fluvial or adfluvial) or resident. We define the fluvial and adfluvial life stages as those individuals that move to exploit growth opportunities outside their natal streams and then return to spawn, whereas resident individuals remain in their natal streams. In this study fluvial refers to individuals that move to the mainstem Stehekin River, whereas adfluvial refers to individuals that migrate to Lake Chelan. Consideration of the expression of the multiple migratory life histories is important in a demographic model because the assumption is that migratory individuals will be larger and more fecund [27].

We used a life-stage based demographic matrix model for bull trout that was similar to those used in previous bull trout reintroduction decision support efforts [16,17]. Six life stages were considered to represent resident and migratory life histories. These stages were fry, juveniles, resident subadults, resident adults, fluvial subadults, fluvial adults, adfluvial subadults, and adfluvial adults (Fig 2). For each of the recipient stream $j$ we used the same transition matrix ($A$):

$$A_j = \begin{pmatrix} 0 & 0 & 0 & F_4*B_j & 0 & F_6*B_j & 0 & F_8*B_j \\ G_1*B_j & P_2*B_j & 0 & 0 & 0 & 0 & 0 & 0 \\ 0 & r_j*G_{2r}*B_j & P_3*B_j & 0 & 0 & 0 & 0 & 0 \\ 0 & 0 & G_3 & P_4 & 0 & 0 & 0 & 0 \\ 0 & f_j*G_{2f}*B_j & 0 & 0 & P_5 & 0 & 0 & 0 \\ 0 & 0 & 0 & 0 & G_5 & P_6 & 0 & 0 \\ 0 & a_j*G_{2a}*L & 0 & 0 & 0 & 0 & P_7*L & 0 \\ 0 & 0 & 0 & 0 & 0 & 0 & G_7*L & P_8 \end{pmatrix}$$

where $F_i$ is the fecundity of stage $i$, $G_i$ is the probability of surviving and transitioning to stage $i+1$, and $P_i$ is the probability of surviving and staying in stage $i$ (S1 Table). Fecundity for resident ($F_4$), fluvial ($F_6$), and adfluvial ($F_8$) adults was the product of the average number of eggs per female in stage $i$, proportion of females in stage $i$, survival of eggs, and the probability of spawning in stage $i$. Thus, the product calculated was the total number of fry emerging from redds. We accounted for the consequences of nonnative brook trout (B) in each stream $j$ and lake trout (L) by applying a penalty to stage-based

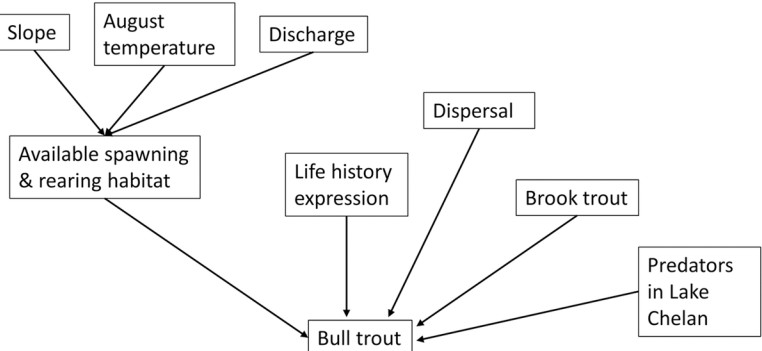

**Fig 2. Influence diagram for bull trout in the Lake Chelan watershed.** Lines pointing to bull trout influence parameters like survival or fecundity in the stage-based demographics. See text for details on each parameter and how they influence bull trout.

survival based on partners' expert opinion (see below for more detail). The proportion of juvenile fish that can transition to the resident (*r*), fluvial (*f*) or adfluvial (*a*) subadult was estimated by partners based on their knowledge of presence of barriers to upstream movement by bull trout and local knowledge of each stream *j* (Table 1). We assumed an individual expressed a given migratory life history for the remainder of its life; however, for this model, their progeny can express any life history. This assumption aligns with the expectation that migratory life history expression in bull trout is at least in part a facultative response to environmental variability [28,29]. Initial parameter values for survival were from a detailed study of bull trout populations in the Walla Walla River, WA [30], and then modified by partners based on their best professional opinions regarding bull trout in the Lake Chelan watershed (S1 Table).

Vital rates in ***A*** were modified by threats identified by partners, which include stream-specific consequences of available habitat and density dependence, climate change, introduced brook trout ($B_j$), predators in Lake Chelan (L), and

**Table 1. Characteristics for each recipient stream including spawning and rearing habitat (stream length) under current (<=year 20) and future (>year 20) conditions, the presence (1) or absence (0) of nonnative brook trout, the proportion of the population that could exhibit adfluvial (*a*), fluvial (*f*) and resident (*r*) life history expression, and the proportion and destination of dispersing fish that could occupy downstream habitat. Life history expression and dispersal were based on partners expert opinion. Future spawning and rearing habitat estimates were based on temperature and discharge projections (see text for details).**

| Recipient stream | Spawning/rearing habitat (km) | | Brook trout presence | Proportion of life history expression | | | Dispersal |
|---|---|---|---|---|---|---|---|
| | Current | Future | | Adfluvial | Fluvial | Resident | |
| Agnes | 55.7 | 29.8 | 0 | 0 | 0 | 1 | Lower Stehekin |
| Boulder | 9.3 | 5.8 | 0 | 0 | 0 | 1 | Lower Stehekin |
| Bridge | 39.5 | 12.6 | 0 | 0 | 0.5 | 0.5 | Mid-Stehekin (0.8); Lower Stehekin (0.2) |
| Company | 19.8 | 12.2 | 0 | 0 | 0 | 1 | Lower Stehekin |
| Fish | 9.3 | 1.2 | 0 | 0.25 | 0 | 0.75 | Lower Stehekin |
| Mid-Stehekin | 34.0 | 7.4 | 0 | 0 | 0.5 | 0.5 | Lower Stehekin (0.6); Bridge (0.4) |
| Prince | 21.3 | 7.6 | 0 | 0 | 0 | 1 | Lower Stehekin (0.5); Fish (0.5) |
| Rainbow | 8.7 | 5.2 | 0 | 0 | 0 | 1 | Lower Stehekin |
| Up-Stehekin | 13.8 | 2.6 | 0 | 0 | 0 | 1 | Mid -Stehekin (0.9); Bridge (0.1) |
| Lower Stehekin | 34.0 | 5.2 | 1 | 0.75 | 0.1 | 0.15 | Fish |
| Park | 9.3 | 5.6 | 0 | 0 | 0 | 1 | Mid -Stehekin (0.5); Bridge (0.5) |
| NF Bridge | 15.7 | 10.2 | 0 | 0 | 0 | 1 | Mid -Stehekin (0.9); Bridge (0.1) |

dispersal. A density dependent relationship was used to calculate the survival and transition of fry to juvenile bull trout ($G_1$) [16,27]:

$$G_1 = DI * \left\{ 1 - exp \left[ \frac{-(K * habitat_j)}{DI * \sum N_j} \right] \right\}$$

where $DI$ is the density independent survival of fry, $K$ is the carrying capacity of juveniles per river kilometer of suitable spawning and rearing habitat in stream $j$ ($habitat_j$), and $N_j$ is the abundance of juveniles in stream $j$ (Table 1; S1 Table). Parameter values for the density dependent equation were taken from previous modeling efforts [16,17] and then modified by partners based on regional demographic data from similar systems [31,32].

The length (km) of suitable spawning and rearing habitat of bull trout was determined from estimates of the potential habitat owing to species-specific habitat characteristics. For this exercise, we used cut-off metrics for gradient (<15%; [33]), discharge (>0.034 m³/s; [34]) and temperature (<12.6 C; [21]) from high resolution National Hydrography Dataset streamlines [35]. Stream sections with gradients greater than 15% were removed from consideration because the probability of bull trout occurring in these sections is 20% or less [33]. Stream sections with a mean summer discharge less than 0.023 m³/s were removed under the assumption that these streams would be less than 2 m wide and unlikely to be used by bull trout [34]. Gradients and summer discharge were estimated from the Beaver Restoration Assessment Tool (BRAT; [36]). Streams sections with a summer mean temperature less than 12.6 C [21] were considered suitable for bull trout spawning and rearing (i.e., stream sections with mean summer temperature > 12.6 C were eliminated; [34]). We estimated habitat length under current conditions and future conditions using year 2040 for changes in temperature [21] (Table 1). We assumed at year 20 in the model that any habitat changes would immediately occur. For example, available habitat in the lower Stehekin River was 21.3 km under current conditions and was reduced at year 20 to 7.6 km because estimated temperatures exceeded 12.6 C.

Multiple species that have been introduced to the Lake Chelan watershed may negatively impact reintroduction efforts. Introduced brook trout can have negative effects on bull trout through hybridization, competition, or predation [11]. When brook trout were present in a stream, partners chose to apply a penalty of 10% reduction ($B_j$) to survival of each egg, fry, juvenile and subadult bull trout. We assumed this reduction would collectively account for potential consequences of hybridization, predation, and competition. This value was set at a low relative level by participating experts due to current low brook trout densities within the system coupled with low quantities of suitable brook trout habitat within areas where were they could occur. Environmental DNA sampling confirmed that brook trout are only present in the lower Stehekin River [25]. Lake trout and other predators in Lake Chelan may reduce the abundance of adfluvial bull trout [37]. Partners assumed a 20% reduction in adfluvial subadult and adult survival would occur to account for predation of this early migratory life. Reduction in survival owing to lake predators were based on expert opinion of the partners.

Bull trout can exhibit a metapopulation or similar dynamic where they can disperse and colonize or recolonize available habitat [28,38,39]. Moreover, dispersal is a common observation for most life stages of bull trout in the region [40,41]. In adopting this view of the spatial population dynamics of bull trout, partners specified that dispersal would be high (20% of reintroduced fish) during the years of reintroduction. During reintroduction, 20% of the individuals that survive would disperse to adjacent streams and incur a survival penalty of 20%. Here, we assumed all life stages, except eggs, would disperse. In subsequent years following reintroductions, 2% of the population in a stream were assumed to disperse [42]. We assumed bull trout could disperse past barriers in the downstream direction but not upstream. For example, individuals reintroduced to Agnes Creek could disperse downstream past the upstream movement barrier into the lower Stehekin River but could not disperse upstream from the lower Stehekin River into Agnes Creek. The proportion of the dispersing bull trout that could disperse to each recipient stream was based on partners judgment (Table 1). For example, if 10 individuals dispersed from Bridge Creek, 8 would go to the mid-Stehekin River and 2 would go to lower Stehekin River.

## Model simulations and sensitivity

For each alternative considered, individuals were reintroduced for the first five years; starting at year 1 and ending at year 5. A reintroduction survival penalty of 20% was applied each year of reintroduction to the life stage being reintroduced [16,43] to account for costs of transport and naiveté to a new habitat. Simulations continued up to 70 years based on partners defined timelines. We present results from each of the five points in time (henceforth year) that partners identified (5, 10, 30, 50, 70), which incorporate the ideas of early monitoring, bull trout generations, and climate change.

Stochasticity was applied in two ways, demographic and environmental. Demographic stochasticity was assumed to account for annual variation in bull trout abundance owing to endogenous (demographic) and exogenous (environmental) factors not considered specifically within the model. After population projections for each year, we used a binomial distribution to select the number of individuals surviving each year drawn from the number of individuals in life stage $i$ and their survival parameter as the probability. Environmental stochasticity was considered in terms of rare, extreme disturbances (e.g., high intensity wildfire, flooding, disease) that could negatively impact bull trout. We assumed an event like this would have an annual probability of 0.05 using random draws from a binomial distribution within the model. If disturbance did occur, abundances of all life stages were reduced by 50%.

We first provided partners with the stochastic results from the range of number of individuals within each life stage. Based on these results, partners identified three potential values for the number of individuals to add that may be plausible. For subadults and adults, partners chose the addition of 30, 60 and 100 individuals per year for the first 5 years. For the juveniles, it was the addition of 200, 500, and 2000; and for eggs, 5,000, 10,000 and 20,000 per year for the first 5 years. We conducted sensitivity analyses on the subset of partner-specified reintroductions for each year.

We evaluated sensitivity to the model parameters using a global sensitivity analysis and to decisions using one-way sensitivity. For the global sensitivity analysis, a Latin hypercube sampling was used to simultaneously vary all parameters [44]. This sampling approach divides each parameter into N bins, where N is the number of simulations, and a random parameter value is selected within each bin for each parameter. Bin combinations are then randomly assigned, which ensures adequate sampling across the range of each parameter. Random forests [45] were then used to rank each parameter by importance to the number of adult bull trout for each recipient streams and year. For parameters between 0 and 1 (e.g., $P_i$, $G_i$, L), a beta distribution was used and a gamma distribution was used for parameters greater than 1 (e.g., eggs per female, K). In the beta and gamma distributions, we used the literature values (S1 Table) for the mean and a c.v. of 0.3. For length of available spawning and rearing habitat, a multiplier (range: 0.5–1.5) was used with a uniform distribution to vary stream-specific habitat length (Table 1) where 0.5 suggests a 50% decrease in habitat, 1.5 is a 50% increase, and 1.0 is no change in habitat. In addition, we varied the number of consecutive years for a reintroduction ranging from 1–5 years to explore the importance of the freT¶uence of consecutive reintroductions.

To explore the effect of model parameters on decision outcomes, a one-way sensitivity was used where all values are held at their literature value (S1 Table) and one parameter is adjusted using the same distribution described above. The one-way sensitivity was done for each parameter in the model. For the purposes of this manuscript, we present results for the addition of 60 adult bull trout for five consecutive years at Year 10 and 30 for subset of parameters that were identified as being sensitive in the global sensitivity analysis.

Model simulations were run in R [46]. We used 500 simulations to account for stochasticity. Results for all scenarios can be found Benjamin and Dunham [47].

## Results

### Optimal decisions

The most optimal decision depended on the year of interest, life stage, and number of individuals added to a recipient stream (Fig 3). Overall, simulations most frequently indicated reintroductions into the lower Stehekin River or Agnes Creek

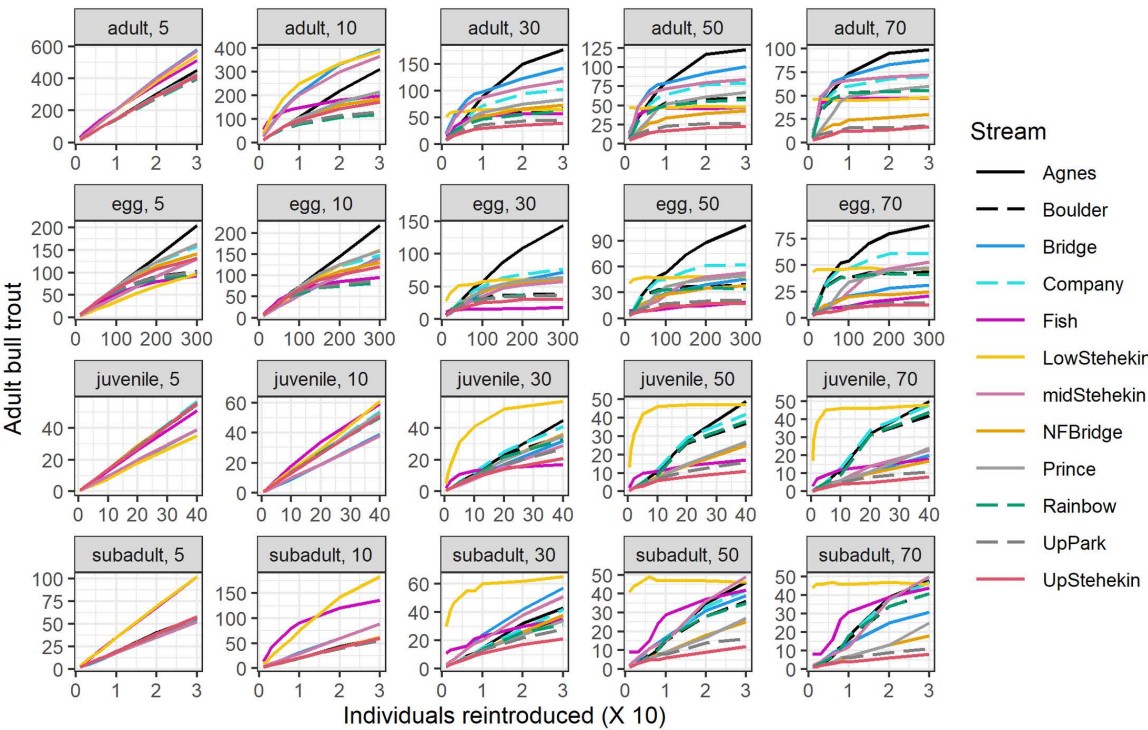

**Fig 3. The number of individuals in a life stage (adult, egg, juvenile, subadult; across rows) reintroduced to a recipient stream (colored lines) and the number of adult bull trout in simulations of a given year (5, 10, 30, 50, 70; across columns).** Reintroductions occurred in each of the years 1-5. Note the scale of the y-axis changes among plots.

produced the maximum number of adult bull trout, the primary fundamental objective. Other recipient streams that consistently produced higher number of adult bull trout across all reintroduced life stages were Company, Boulder, and Rainbow Creeks. Park Creek and the upper Stehekin River consistently produced the lowest number of bull trout across all reintroduced life stages and years. Responses in other recipient streams varied depending on the life stage and year. For example, Bridge Creek and mid-Stehekin River produced a larger number of adults when the life stage reintroduced was adults, and a modest number of adults when juveniles were reintroduced.

Model results indicated that by year 30 a consistent point of diminishing returns (limited increases in abundance of adult bull trout relative to introduced individuals) was realized among most of the recipient streams and migratory life histories (Fig 3). This point was approximately 50–100 individual subadults and adults that were added into most streams. For eggs, it was approximately 5,000–10,000 individuals added and for juveniles was approximately 2,000 individuals. During years 5 and 10, the number of adults increased with the addition of individuals from each life stage. At year 30–70, the abundance of adult bull trout was consistent for all numbers of individuals reintroduced within a life stage.

Increases in the number of adult bull trout declined over time in all recipient streams (Fig 3). At year 5, the number of adults increased with more individuals added regardless of life stage. At year 10, declining rates of increase were more evident. By year 30, the number of adults began to increase more slowly, and the population started to stabilize to a consistent size of less than 50 adults for most recipient streams.

In general, presence of a migratory life history expression or more available spawning and rearing habitat produced more adult bull trout over time. This was most evident with the lower Stehekin River that had a large proportion of the population that would migrate to Lake Chelan. Streams that had a higher number of simulated adult bull trout but did not

have a migratory expression such as Agnes, Company, and Boulder Creeks had only resident bull trout that dispersed to the lower Stehekin River. Some of the progeny of resident adults became adfluvial and indirectly increased abundance identified for a recipient stream (e.g., Fig 4).

## Sensitivity

We present results of the global sensitivity analysis for reintroduction of 60 adults at years 5–70 for Agnes Creek and lower Stehekin River. In general, patterns of sensitivity were similar for all the reintroduction scenarios for life-stage and individuals reintroduced and years to realize the fundamental objective. Results from additional sensitivity analyses can be found in an associated data release [47].

Global sensitivity analysis suggested differences in the most sensitive parameters among recipient streams and years (Fig 5). For the lower Stehekin River, the most sensitive parameters were related to adfluvial migratory life history expression. Specifically, the three most important parameters were predation in Lake Chelan ($L$), survival of adfluvial adults ($G_7$, $P_8$), and survival from juvenile to adfluvial subadult ($a.G_2$). In contrast, the most sensitive parameters in Agnes Creek, and other recipient streams that were solely residents (e.g., Company and Rainbow Creeks), were survival from juvenile to resident subadult ($r.G_2$), survival from resident subadult to resident adult ($G_3$), and the egg survival ($G_0$). Parameter sensitivity was consistent among streams with similar migratory life history expression (e.g., all fluvial recipient streams; [47]). Although the effects of brook trout on bull trout are a common concern, the discount in survival to bull trout life stages owing to brook trout ($B$) was of moderate importance relative to other parameters.

For the one-way sensitivity analysis, we present parameters that were identified as most sensitive in the global sensitivity analysis (i.e., $a.G_2$, $f.G_2$, $G_0$, $L$, and $r.G_2$; Fig 6). In years 5 and 10, one-way sensitivity analysis suggested the lower

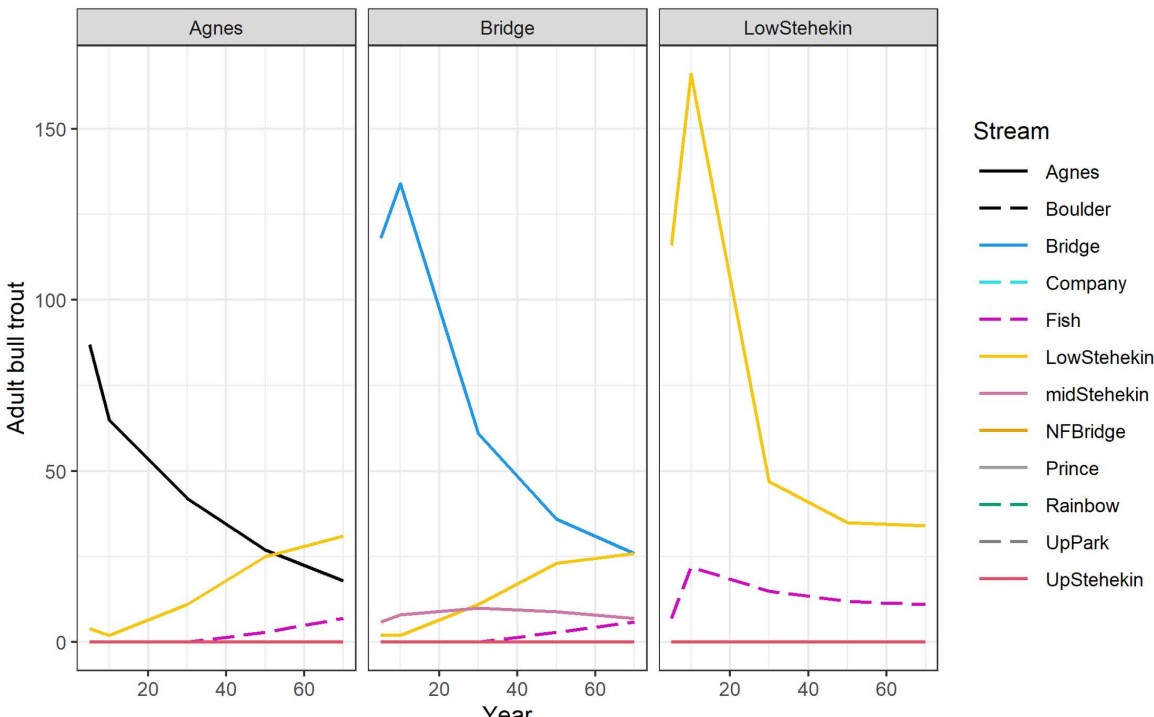

**Fig 4. Example of dispersal of bull trout from representative recipient streams to an adjacent stream under the scenario of the reintroduction of 60 adults/year simulated at year 30.** Reintroductions occurred in years 1-5.

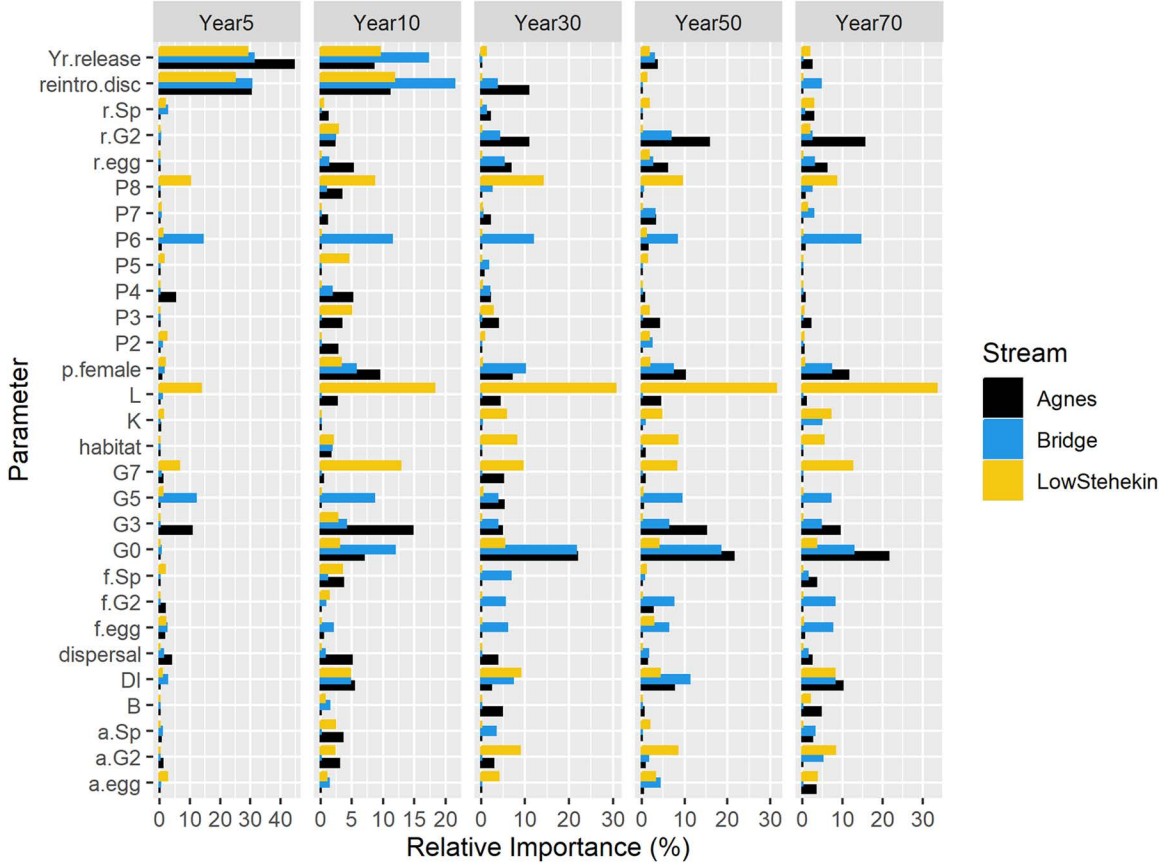

**Fig 5. Global sensitivity analysis for the reintroduction of 60 adults into two recipient streams (Agnes Creek, lower Stehekin River) that were consistently optimal decisions.** The larger the bars the more sensitive the fundamental objective (i.e., number of adult bull trout) is to the parameter. Patterns are also similar for other recipient streams with a similar life history expression [see 47]. *reintro.disc* = Reintroduction survival discount; *p.female* = Proportion of females; *f.Sp* = Annual spawning probability of fluvial adults; *r.Sp* = Annual spawning probability of resident adults; *a.Sp* = Annual spawning probability of adfluvial adults; *DI* = maximum fry survival; *G0* = Egg to fry survival; *G3* = Resident subadult to adult survival; *G5* = Fluvial subadult to adult survival; *f.G2* = Juvenile to fluvial subadult survival; *a.G2* = Juvenile to adfluvial subadult survival; *f.egg* = Eggs per fluvial adult female; *a.egg* = Eggs per adfluvial adult female; *r.egg* = Eggs per resident adult female; *P2* = Survival and persistence as a juvenile; *P3* = Survival and persistence as a resident subadult; *P4* = Survival and persistence as a resident adult; *P5* = Survival and persistence as a fluvial subadult; *P6* = Survival and persistence as a fluvial adult; *r.G2* = Juvenile to resident subadult survival; *P7* = Survival and persistence as an adfluvial subadult; *G7* = dfluvial subadult to adult survival; *P8* = Survival and persistence as a adfluvial adult; *K* = Juvenile carrying capacity per river kilometer; *L* = Lake Chelan predator discount; *Yr. release* = Consecutive years released; *B* = Brook trout discount. See main text and S1 Table for description of parameter abbreviations and values.

Stehekin River was the most optimal recipient stream to reintroduce bull trout. However, the optimal recipient stream shifted to other recipient streams in later years (30–70). For example, in year 30, low values of $a.G_2$ and $L$ suggested Bridge Creek would be optimal for reintroductions of 60 adult bull trout, whereas at high values for these parameters, the mid-Stehekin River was most optimal. In contrast, at low values for $G_0$ and $r.G_2$ suggested the lower Stehekin River was optimal, Agnes Creek was optimal at high values for $G_0$ and $r.G_2$, and Bridge Creek at intermediate values.

## Discussion

Assessments of species reintroductions can benefit from rigorous feasibility assessments to evaluate the consequences of decision alternatives [2]. Here we report a new example of a reintroduction feasibility assessment for bull trout, a species that has been the subject of several recent studies focused on the feasibility of reintroductions or other types of

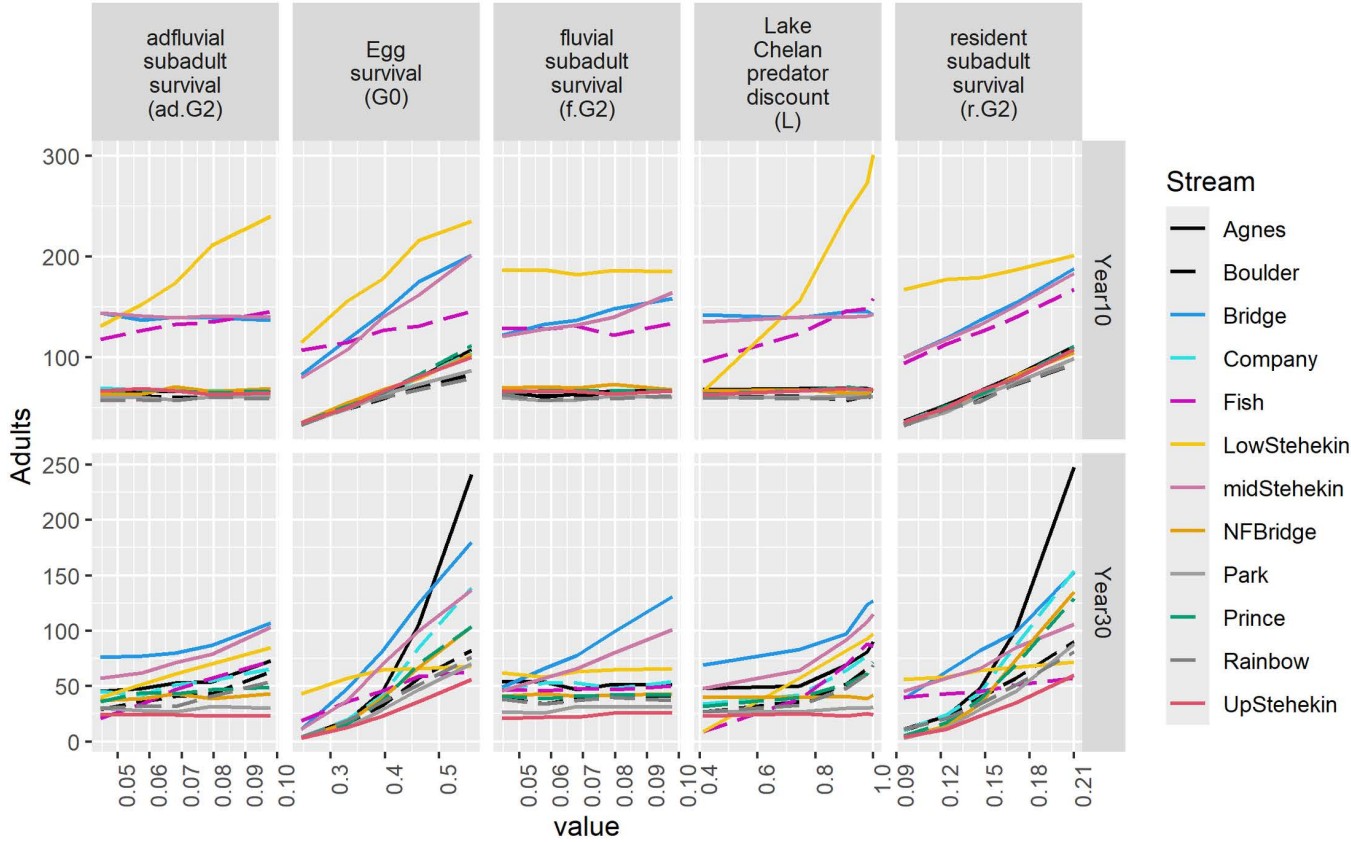

**Fig 6. One-way sensitivity response profiles of most sensitive parameters for abundance in year 10 and 30 when reintroducing 60 adults/year for five years.** Lake Chelan predator discount ($L$) of 1 is predators have no effect on bull trout and of 0 is removes all bull trout in the lake.

translocations [14–19,48]. Although in some cases, past reintroductions have yet to produce desirable outcomes [49], actions resulting from more rigorous feasibility assessments may hold greater promise. Furthermore, involving partners in processes such as structured decision making and co-production of models as implemented herein arguably results in more positive social outcomes from the start of any process to conduct a reintroduction [50]. The case study of the Lake Chelan basin provides new insights to a growing body of work on bull trout reintroductions, which we discuss in more detail below.

## Reintroduction strategies

Potential reintroduction strategies for bull trout can range widely. For strategies considered herein, results indicate that outcomes are sensitive to nuances of a reintroduction strategy that include the life stage considered, number and survival of individuals added, and the frequency of additions. The release of adults had the greatest utility in that the subsequent abundance of reintroduced adults produced in any recipient habitat was higher compared to the releases of other life stages. This is likely owing to the assumption in the model of spawning occurring during the first year for most of the added individuals, where one adult can produce 625–4320 eggs depending on individual body size, which is associated with migratory life history expression.

The potential benefits of relying on adult bull trout for a reintroduction are clear from a modeling perspective, but in practice logistical concerns, such as the feasibility of capturing, transporting and releasing a given life stage, and risks to

donor populations will be important considerations. For example, roadless or wilderness areas that are prevalent within the upper Stehekin River and tributaries may be more conducive to using hatch boxes [51] because eggs may be easier to transport. Regardless of life stage, there was a decreasing marginal benefit as more individuals were reintroduced to a stream. In part, this is contingent on our assumptions of the density dependence equation used, a limitation of all population models that we further discuss below.

In this study, we did not consider specific reintroduction strategies such as artificial propagation, translocation, or hatch boxes [17,43], each of which can have their own penalties on survival. Instead, we varied the reintroduction penalty in the sensitivity analyses as a generic surrogate for different strategies. Simulations suggest that this reintroduction penalty will be highly important during year 5 and persist up to year 10, but not in subsequent years. This is a sensible finding, as internal population dynamics in the established population of reintroduced individuals should be more important over time.

## Connectivity of recipient streams

Connectivity is an important theme for restoration of salmon and trout populations [29]. One potential benefit of increased connectivity for bull trout is access to habitats that allow individuals to attain much larger sizes than would be possible in their natal habitats [27]. Results from this feasibility assessment of a bull trout reintroduction indicate that establishing a population that can support migratory individuals may be more successful. This is because the migratory females are often larger in size and more fecund relative to resident females [52]. The greater fecundity of adfluvial adult females, such as in the lower Stehekin River, could offset the negative impacts of introduced brook trout in the recipient stream and predators in Lake Chelan. A similar modeling approach considering management alternatives for minimizing impacts of introduced brook trout on bull trout reinforces this finding [48]. Partners believed connectivity and dispersal could be important for bull trout in the Lake Chelan basin. As such, we included a dispersal parameter where a high proportion (0.2) of individuals dispersed during reintroductions and then a lower proportion (0.02) in subsequent years. Although the dispersal parameter was not identified as being sensitive in the model or decision, other parameters suggest it may be more important than results suggest. For example, streams lacking migratory connectivity (e.g., Agnes, Company, Boulder) exhibited increasing sensitivity over time to parameters associated with an adfluvial life history expression (i.e., predators in Lake Chelan). Findings of this modeling effort align with observations in natural populations where dispersal of bull trout can support downstream populations [53]. We can only speculate how important this dynamic was in the Stehekin River, but reports of large numbers of fish accumulating below barriers during times when adults should be migrating upstream [13] suggest it could have been the case. In any case, this dynamic was important within the model employed herein.

## Additional considerations

One fundamental objective that partners decided not to include in the model was the impact of bull trout on other fish, such as westslope cutthroat trout, rainbow trout, Chinook salmon, kokanee, and pygmy whitefish. Bull trout can consume any of these other fish of concern, depending on sizes they attain [54]. Consumption of these native species within the Stehekin River, Lake Chelan, or even downstream in the Chelan and Columbia Rivers (if individuals leave the system) are concerns. Further consideration of these potential impacts may be warranted, as has been the case for other reintroductions [19,55]. If a reintroduction is attempted, additional study could provide further insight to interactions among these species, which can be complex [54,56].

## Limitations and future directions

As with all models, the one presented here has limits. Foremost in our interpretation, leveling off the adult numbers over time could be owing to carrying capacity reached during the initial years when individuals were added. Density dependence is often a strong influence in demographic models like those used here [57,58]. The influence of density dependence may

also be stream-specific or occur on smaller spatial scales [57] and could be considered further. Second, values for all demographic parameters are uncertain for bull trout in the Lake Chelan basin. Most demographic parameters we used were from the Walla Walla River, WA [30] that may not be representative of reintroduced fish in Lake Chelan tributaries. Given that it is unlikely these uncertainties can be fully resolved, we attempted to evaluate their consequences in a series of sensitivity analyses. These allowed us to identify important parameters that could be addressed with monitoring if a reintroduction is attempted. For example, in the early years following reintroductions, the survival of reintroduced individuals and number of consecutive years of reintroduction were highly influential on the outcome of the model. Whereas, in later years (≥30) the survival of younger stages, such as eggs, fry and juveniles, were more influential on model outcomes. This prospective look at the changing importance of different parameters could critically inform long-term efforts to track reintroduction success.

Sensitivity analyses could also identify potential management actions. For example, predators in Lake Chelan were highly influential to adfluvial populations in the lower Stehekin River across all years evaluated. A major potential predator of bull trout in Lake Chelan is introduced lake trout. For example, lake trout can consume over 500 kg of kokanee per year in Lake Chelan [59] and could have a strong impact on bull trout entering the lake. Because of the importance of lake predators in the model, it may suggest that suppressing predators, like lake trout, could benefit reintroduced bull trout [37,60]. It is worth noting that, while lake trout competition and predation can lead to bull trout population extirpation, sympatric introduced populations exist and bull trout and lake trout coexist naturally where their natural ranges overlap [61,62], so potential outcomes in Lake Chelan are far from certain. This is another potential priority for future empirical study if a reintroduction is attempted.

## Conclusions

Our assessment of a potential reintroduction of bull trout to the Lake Chelan basin provides a unique example highlighting the context-dependent considerations that can be effectively addressed in considering the feasibility of a reintroduction. Relative to other cases studied to date, the Lake Chelan basin presents some unique potential threats (e.g., introduced lake trout), unique value to migratory life history expression and connectivity, and as always, several critical uncertainties regarding assumptions within the model applied. What is clear, however, it that environmental conditions (i.e., widespread availability of cold water) within the Lake Chelan system offer bull trout the potential to establish a population that may be more likely to persist in the face of warming climates, even relative to many extant populations [21]. As such, the Lake Chelan system offers a number of important challenges and opportunities to weigh in considering the value of a potential reintroduction to benefit recovery of bull trout and to establish a population that may persist long into the future.

## Supporting information

**S1 File. Bull trout demographic parameter values.**
(DOCX)

## Acknowledgments

We thank the following partners, along with some of the authors of this paper, for supporting and participating in the workshops for model development: Jeremy Cram, Stephen Caromile, and Travis Maitland (Washington Department of Fish and Wildlife); Bret Nine (Confederate Tribes of the Colville Reservation); Madeleine Eckmann, Jeff Caisman, and Dave Blodgett (Yakama Nation); Bill Gale, and R.D. Nelle (U.S. Fish and Wildlife Service); Tracy Bowerman (Upper Columbia Salmon Recovery Board); Mike Kaputa (Chelan County Natural Resource Department); Bill Towey and Scott Hopkins (Chelan P.U.D.); Mariah Mayfield (Okanogan-Wenatchee National Forest. We thank Tracy Bowerman for assistance with spawning and rearing habitat estimates. Sophia Wagner (Student Conservation Association) assisted manuscript preparation. Dylan Gomes, Mark Sorel, Amber Steed, and Brett van Poorten provided comments that greatly improved this

manuscript. Any use of trade, firm, or product names is for descriptive purposes only and does not imply endorsement by the U.S. Government.

## Author contributions

**Conceptualization:** Joseph R. Benjamin, Judith Neibauer, Ashley Rawhouser, Jason B. Dunham.

**Data curation:** Joseph R. Benjamin, Judith Neibauer, Hugh Anthony, Jose Vazquez.

**Formal analysis:** Joseph R. Benjamin.

**Funding acquisition:** Joseph R. Benjamin, Judith Neibauer, Ashley Rawhouser, Jason B. Dunham.

**Methodology:** Joseph R. Benjamin, Hugh Anthony, Jose Vazquez, Ashley Rawhouser, Jason B. Dunham.

**Validation:** Joseph R. Benjamin, Hugh Anthony, Jose Vazquez.

**Visualization:** Joseph R. Benjamin.

**Writing – original draft:** Joseph R. Benjamin, Judith Neibauer, Hugh Anthony, Jose Vazquez, Jason B. Dunham.

**Writing – review & editing:** Joseph R. Benjamin, Judith Neibauer, Hugh Anthony, Jose Vazquez, Ashley Rawhouser, Jason B. Dunham.

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
