## [Decision Letter · Decision Letter 0]

25 Feb 2025

PONE-D-25-03684Decision support model for the feasibility of bull trout reintroductionsPLOS ONE

Dear Dr. Benjamin,

Thank you for submitting your manuscript to PLOS ONE. After careful consideration, we feel that it has merit but does not fully meet PLOS ONE’s publication criteria as it currently stands. Therefore, we invite you to submit a revised version of the manuscript that addresses the points raised during the review process.

We have received two thorough reviews, both providing clear recommendations how to improve the manuscript. Please adress the comments of the reviewers in your revision.

We look forward to receiving your revised manuscript.

Kind regards,

Florian Borgwardt

Academic Editor

PLOS ONE

Journal Requirements:

Additional Editor Comments:

I have received two thorough reviews for your manuscript. Both reviewers give clear recommendations how to improve the manuscript. Please address the comments in your revision.

Reviewers' comments:

Reviewer's Responses to Questions

**Comments to the Author**

1. Is the manuscript technically sound, and do the data support the conclusions?

Reviewer #1: Yes

Reviewer #2: Yes

2. Has the statistical analysis been performed appropriately and rigorously? 

Reviewer #1: Yes

Reviewer #2: Yes

3. Have the authors made all data underlying the findings in their manuscript fully available?

Reviewer #1: No

Reviewer #2: Yes

4. Is the manuscript presented in an intelligible fashion and written in standard English?

Reviewer #1: Yes

Reviewer #2: Yes

5. Review Comments to the Author

Reviewer #1: Review of "Decision support model for the feasibility of bull trout reintroductions"

It was my pleasure to review this manuscript. I think the ideas behind this paper, the ways in which the authors fulfilled the study and the writing overall were very well done and quite thorough. As such, I have few comments on the manuscript provided. I would note that I was unable to access the supplemental material, and so cannot state whether all data and code were provided with the manuscript.

I like the layout of the structured decision-making approach and especially the creation of the problem statement, which then focused all future discussions. Overall, I really like presenting this paper as a co-produced product to inform group decisions, and think more work should be done like this. As an aside, I think the authors limited the scope of this project to some extent. The title alone "decision support model..." makes the paper seem much narrower than it was. This paper is not just about the model, but its potential to inform reintroduction. I recognize that this is the rest of the title, but I think a lot more could have been written about the co-production process and how members viewed the model and how this may have informed their decisions going forward. Perhaps this is a follow-up paper, but I found I was a bit disappointed that results were so limited to this model, when the value of it is the co-production process and the potential to really help all participants embrace findings and work to act on its advice.

In the introduction, the last two paragraphs have significant overlap; I initially read both as an objectives paragraph. If it were possible to keep the second-to-last paragraph about the study system and the last paragraph to an explicit discussion of what the paper aims to do, that would help clarify interpretation. Even just changing the wording on line 73, so the topic sentence is broader about a study system, rather than what the paper is reporting on, would help in that regard.

The transition matrix could be more clearly described. I found myself going back and forth through two different paragraphs to try and determine what I was looking at. I would suggest: (1) clearly describing columns/rows near line 180, rather than the top of that paragraph, so it is easier to envision what each column represents; (2) briefly describe what B[j] and L are in the paragraph with the transition matrix, and then more fully describing them in the next paragraph (which is what was done); (3) give values and descriptions of r[j], f[j] and a[j] in the paragraph on line 180. I still don't know what those values might be and the terms themselves are only indirectly identified.

The recruitment function on line 203 is interesting and packs a lot in. I would expect this over-compensatory structure to lead to a lot of oscillations in abundance within each simulation, relative to a more typical Beverton-Holt structure (though I think a Ricker function, as is presented, is probably more realistic for such an aggressive species). Are findings sensitive to the choice of recruitment function? With all the sensitivity analysis, this is one area that wasn't touched on.

That's it! I really like this paper and appreciate the time that went into writing it so well.

Reviewer #2: Overall, this paper has value to the bull trout conservation community, particularly those considering reintroductions and translocations. I appreciate the application of structured decision making and sensitivity analysis to this challenging topic with relatively few case studies to learn from. As someone working in this space, examples of how to navigate the decision process are helpful. I have included questions and clarifying comments below but recommend publication with some minor revisions.

Minor grammatical issues to address.

Consider clearly stating that structured decision-making was used in the abstract, given that it was central to the paper.

Line 149: Note time duration of donor infusion within this paragraph. I see this is addressed later in line 165+, but it leaves the reader wondering about this important metric until they reach that point.

I would like to see discussion of all the causes that led to extirpation in the first place to assure stakeholders that these have been adequately addressed and threats to persistence are removed.

Line 189, describe why you assumed progeny could express any life history when we don’t know that to be the case. Citations here would support your choice.

Line 204, how was carrying capacity determined per river kilometer? I see consideration of factors, but not how you reached a designated K.

It appears that habitat suitability relies on presence of suitable temperatures, stream discharge, and sufficient gradient. I assume also that bull trout previously existed in these reaches and therefore could again, but is that the case everywhere being considered? You may have covered this but it could be reiterated here for clarity.

Line 226, how was the 10% reduction in survival determined? Further, you may state this but could make clear how sensitive the analyses was to this rate (i.e., if it's truly much different, would predictions change).

Line 230, how was a 20% reduction in survival selected as a consequence of lake trout presence in the system?

Line 236, please clarify how the 20% dispersal rate was selected.

Line 263, how was abundance of all life stages reduced by 50%, is there a citation or reasoning associated with this value?

Line 267 through 270, I assume these values refer to additions per year?

Line 406, define translocation as compared to what’s being proposed in this study.

Table 1, consider including in caption that future habitat availability estimates are based on temperature projections.

Figure 4, does this refer to the introduction of 60 adults total over 5 years? If so, consider clarifying that for the reader, including number of fish/year.

Figure 5, could parameters included be described in a way in the figure such that readers don't need to consult a supplemental table? This is useful information, but fairly busy and difficult to read. I suggest some reworking to make more readable and not requiring a supplemental table to understand.

It's interesting that the Lower Stehekin River predicted better long term outcomes than other sites, despite the presence of nonnatives in the river and connected lake system. I assume this is related to system productivity and predicted increases in adfluvial fish fecundity. I think it's worth more discussion of this point for managers weighing trade-offs of reintroductions/translocations in systems where removal of nonnatives isn't a viable option.

6. PLOS authors have the option to publish the peer review history of their article (what does this mean? ). If published, this will include your full peer review and any attached files.

**Do you want your identity to be public for this peer review?** For information about this choice, including consent withdrawal, please see our Privacy Policy .

Reviewer #1: **Yes: ** Brett van Poorten

Reviewer #2: **Yes: ** Amber Steed

---

## [Author Response · Author response to Decision Letter 0]

20 Mar 2025

Additional Editor Comments:

I have received two thorough reviews for your manuscript. Both reviewers give clear recommendations how to improve the manuscript. Please address the comments in your revision.

Response: Please see our reconciliations to reviewer comments below.

Reviewer #1: Review of "Decision support model for the feasibility of bull trout reintroductions"

It was my pleasure to review this manuscript. I think the ideas behind this paper, the ways in which the authors fulfilled the study and the writing overall were very well done and quite thorough. As such, I have few comments on the manuscript provided. I would note that I was unable to access the supplemental material, and so cannot state whether all data and code were provided with the manuscript.

Response: Thank you. Sorry about the supplemental material.

I like the layout of the structured decision-making approach and especially the creation of the problem statement, which then focused all future discussions. Overall, I really like presenting this paper as a co-produced product to inform group decisions, and think more work should be done like this. As an aside, I think the authors limited the scope of this project to some extent. The title alone "decision support model..." makes the paper seem much narrower than it was. This paper is not just about the model, but its potential to inform reintroduction. I recognize that this is the rest of the title, but I think a lot more could have been written about the co-production process and how members viewed the model and how this may have informed their decisions going forward. Perhaps this is a follow-up paper, but I found I was a bit disappointed that results were so limited to this model, when the value of it is the co-production process and the potential to really help all participants embrace findings and work to act on its advice.

Response: We agree that the title should mention that this effort was a participatory process. As such we added “Partner-driven” to the beginning of the title as well as made other minor changes.

We also agree that it would be informative to evaluate participants perspective of the structured decision making process and how it informed future decisions. To effectively evaluate perspectives of participant, it would require more of a social science component that would include a questionnaires and interviews. This was beyond the scope of the study we present but definitely a research idea we are interested in pursuing. For decisions going forward, what we present is a first step at ongoing planning and implementation. In other words, it may be too soon to say how this study influenced the final reintroduction decision. Perhaps future research can be done to combine these two points the Reviewer bring up.

In the introduction, the last two paragraphs have significant overlap; I initially read both as an objectives paragraph. If it were possible to keep the second-to-last paragraph about the study system and the last paragraph to an explicit discussion of what the paper aims to do, that would help clarify interpretation. Even just changing the wording on line 73, so the topic sentence is broader about a study system, rather than what the paper is reporting on, would help in that regard.

Response: We modified the first sentence to avoid confusion.

The transition matrix could be more clearly described. I found myself going back and forth through two different paragraphs to try and determine what I was looking at. I would suggest: (1) clearly describing columns/rows near line 180, rather than the top of that paragraph, so it is easier to envision what each column represents; (2) briefly describe what B[j] and L are in the paragraph with the transition matrix, and then more fully describing them in the next paragraph (which is what was done); (3) give values and descriptions of r[j], f[j] and a[j] in the paragraph on line 180. I still don't know what those values might be and the terms themselves are only indirectly identified.

Response: Thank you for the suggestion. We edited the paragraph and added language to clarify the points the Review mentions. Specifically, we (1) moved the description of bull trout life history expression to a new paragraph, which allowed for the description of columns and rows to be closer to the matrix equation; (2) added a sentence to define B[j] and L; and (3) clarified the descriptions of r[j], f[j], and a[j] – the values are recipient stream specific and can be found in Table 1

The recruitment function on line 203 is interesting and packs a lot in. I would expect this over-compensatory structure to lead to a lot of oscillations in abundance within each simulation, relative to a more typical Beverton-Holt structure (though I think a Ricker function, as is presented, is probably more realistic for such an aggressive species). Are findings sensitive to the choice of recruitment function? With all the sensitivity analysis, this is one area that wasn't touched on.

Response: We did not evaluate the sensitivity of the recruitment or density dependent function as a whole. Instead, we included individual parameters, such as habitat size, density independent survival, and carrying capacity, in the sensitivity analyses. Relative to other parameters in the model, those in the density dependent equation had less sensitivity to the outcome of the model. We focused the results on the most sensitive parameters and offer all the results in the data release for readers to assess themselves.

We did not evaluate whether a Beverton-Holt or Ricker type function would be more sensitive. Although beyond the scope of this study, will consider this idea in future efforts. We do discuss the limitation of density depencene in the “Limitations and future directions” subsection of the Discussion.

That's it! I really like this paper and appreciate the time that went into writing it so well.

Response: Thank you!

Reviewer #2: Overall, this paper has value to the bull trout conservation community, particularly those considering reintroductions and translocations. I appreciate the application of structured decision making and sensitivity analysis to this challenging topic with relatively few case studies to learn from. As someone working in this space, examples of how to navigate the decision process are helpful. I have included questions and clarifying comments below but recommend publication with some minor revisions.

Response: Thank you. We address your comments below.

Minor grammatical issues to address.

Response: We are unsure if this is something we need to address.

Consider clearly stating that structured decision-making was used in the abstract, given that it was central to the paper.

Response: Thanks for pointing this out. We agree and added the use of structured decision making in the abstract.

Line 149: Note time duration of donor infusion within this paragraph. I see this is addressed later in line 165+, but it leaves the reader wondering about this important metric until they reach that point.

Response: We agree and clarified this point in the topic sentence of the first paragraph of the “Reintroduction alternatives” subsection of the methods.

I would like to see discussion of all the causes that led to extirpation in the first place to assure stakeholders that these have been adequately addressed and threats to persistence are removed.

Response: There is speculation as to why bull trout were extirpated in the Lake Chelan watershed, but exact causes are uncertain. We identified the three main reasons why extirpation occurred in the introduction in the penultimate paragraph. We do not see a reason to speculate further on these potential drivers. Cited works, including “What happened to Bull Trout in Lake Chelan” by Nelson (2012), address this topic in greater detail.

Line 189, describe why you assumed progeny could express any life history when we don’t know that to be the case. Citations here would support your choice.

Response: In what is now line 207, we added this sentence “. This assumption aligns with the expectation that migratory life history expression in bull trout is at least in part a facultative response to environmental variability [35,49].” Although work on bull trout is not as detailed as for other species of salmonids, facultative expression of migratory life histories is well documented (e.g., Rieman and Dunham 2000; Brenkman et al. 2019). For the reader’s benefit, we considered this to be an assumption here as we do not know for sure in this particular system or how the assumption might apply to the donor population/stock that might be employed, should a reintroduction be attempted.

Brenkman SJ, Peters RJ, Tabor RA, Geffre JJ, Sutton KT. Rapid recolonization and Life history responses of bull trout following dam removal in Washington’s Elwha River. N Am J of Fish Manage. 2019 Jun;39(3):560–73.

Rieman BE, Dunham JB. Metapopulations and salmonids: a synthesis of life history patterns and empirical observations. Ecol Freshw Fish. 2000 Jun;9(1–2):51–64.

Line 204, how was carrying capacity determined per river kilometer? I see consideration of factors, but not how you reached a designated K.

Response: The carrying capacity value was from previous modeling efforts and then modified by partners partners based on regional demographic data from similar systems. We added this statement and citations to the Methods.

It appears that habitat suitability relies on presence of suitable temperatures, stream discharge, and sufficient gradient. I assume also that bull trout previously existed in these reaches and therefore could again, but is that the case everywhere being considered? You may have covered this but it could be reiterated here for clarity.

Response: The amount of habitat available is based on an intrinsic potential analysis. We and the participants cannot say for sure if bull trout actually occupied all the stream length available or not. Because of this uncertainty, we did modify the length of habitat in the sensitivity analysis, which suggested habitat length had moderate importance for the overall outcome of the bull trout reintroduction relative to other parameters considered, like survival of eggs.

Line 226, how was the 10% reduction in survival determined? Further, you may state this but could make clear how sensitive the analyses was to this rate (i.e., if it's truly much different, would predictions change).

Response: The 10% reduction in survival owing to brook trout was based on expert opinion by the participants. We now explicitly state this in the paragraph. We added a sentence specifically stating the model outcomes had moderate sensitivity to the brook trout parameter in the results.

Line 230, how was a 20% reduction in survival selected as a consequence of lake trout presence in the system?

Response: The 20% reduction in survival owing to lake trout was based on expert opinion by the participants. We now explicitly state this in the paragraph.

Line 236, please clarify how the 20% dispersal rate was selected.

Response: The 20% dispersal rate was based on the expert opinion of the partners.

Line 263, how was abundance of all life stages reduced by 50%, is there a citation or reasoning associated with this value?

Response: We used a random binomial function with a probability of 0.05 to determine if a major stochastic event occurred in a given year. If it did, then 50% of the population was reduced. The 50% reduction was based on expert opinion of the partners. We relied on expert opinion for this because there are no studies that have examined survival following rare, extreme disturbances. Accordingly, we could not provide a citation.

Line 267 through 270, I assume these values refer to additions per year?

Response: Good catch. We clarified this.

Line 406, define translocation as compared to what’s being proposed in this study.

Response: We could add a footnote to provide more definition of “translocation” if needed here so it is available within the narrative of this paper. For brevity, we provide citations that readers can refer to.

Table 1, consider including in caption that future habitat availability estimates are based on temperature projections.

Response: Good suggestion. We added this language to the table caption.

Figure 4, does this refer to the introduction of 60 adults total over 5 years? If so, consider clarifying that for the reader, including number of fish/year.

Response: We clarified the caption for the figure to specify it is using the scenario where 60 adults/year were reintroduced for the first 5 years.

Figure 5, could parameters included be described in a way in the figure such that readers don't need to consult a supplemental table? This is useful information, but fairly busy and difficult to read. I suggest some reworking to make more readable and not requiring a supplemental table to understand.

Response: Yes, we agree this Figure has important information and is busy. We were unable to add parameter descriptions in the figures without making it difficult to read. As an alternative, we added the descriptions in the figure caption.

It's interesting that the Lower Stehekin River predicted better long term outcomes than other sites, despite the presence of nonnatives in the river and connected lake system. I assume this is related to system productivity and predicted increases in adfluvial fish fecundity. I think it's worth more discussion of this point for managers weighing trade-offs of reintroductions/translocations in systems where removal of nonnatives isn't a viable option.

Response: We added the following sentences in the “Connectivity of recipient streams” subsection of the discussion “The greater fecundity of adfluvial adult females, such as in the lower Stehekin River, could offset the negative impacts of introduced brook trout in the recipient stream and predators in Lake Chelan. A similar modeling approach considering management alternatives for minimizing impacts of introduced brook trout on bull trout reinforces this finding [45].”

---

## [Editor Report · Decision Letter 1]

8 Apr 2025

A partner-driven decision support model to inform the reintroduction of bull trout

PONE-D-25-03684R1

Dear Dr. Benjamin,

We’re pleased to inform you that your manuscript has been judged scientifically suitable for publication and will be formally accepted for publication once it meets all outstanding technical requirements.

Kind regards,

Florian Borgwardt

Academic Editor

PLOS ONE

Additional Editor Comments (optional):

Dear Authors,

thanks for your revisions. The points raised by the reviewers have been adequately addressed and the paper can be accepted for publication.

Kindest regards
---

## [Editor Report · Acceptance letter]

PONE-D-25-03684R1

PLOS ONE

Dear Dr. Benjamin,

I'm pleased to inform you that your manuscript has been deemed suitable for publication in PLOS ONE. Congratulations! Your manuscript is now being handed over to our production team.

Kind regards,

on behalf of

Dr. Florian Borgwardt

Academic Editor

PLOS ONE